# “Mi Corazón se Partió en Dos”: Transnational Motherhood at the Intersection of Migration and Violence

**DOI:** 10.3390/ijerph192013404

**Published:** 2022-10-17

**Authors:** Laurie Cook Heffron, Karin Wachter, Esmeralda J. Rubalcava Hernandez

**Affiliations:** 1School of Behavioral and Social Sciences, St. Edward’s University, Austin, TX 78704, USA; 2School of Social Work, Watts College of Public Services and Community Solutions, Arizona State University, Phoenix, AZ 85004, USA; 3School of Social Work, University of Texas at Arlington, Arlington, TX 76019, USA

**Keywords:** migration, transnational motherhood, violence against women, intimate partner violence

## Abstract

In the recent Central American migrations spurred by violence, political instability, and economic insecurity, women grapple with whether and when to bring their children with them in pursuit of safety in another country, and with fulfilling their roles as mothers from afar. Drawing from the transnational motherhood literature and critical feminist theories, this interpretive qualitative study examined transnational motherhood grounded in the lived experiences of Central American women (*n* = 19) over the course of their migrations to the US. Informed by the principles of grounded theory, the inductive analysis identified five processes in which migration and violence shaped meanings of motherhood: risking everything, embodying separation, braving reunification, mothering others, and experiencing motherhood due to sexual violence. The findings contribute knowledge of how violence shapes and informs women’s migrations and decision-making, and the consequences women endure in taking action to mitigate threats of violence in their own and their children’s lives. The analysis furthermore highlights the specific and profound effects of family separation on mothers. The voices, perspectives, and experiences of migrating mothers and the ways in which migration and violence shapes notions and lived experiences of motherhood are imperative to research, practice, and advocacy to change oppressive immigration policies.

## 1. Introduction

Against the backdrop of contemporary Central American migrations spurred by violence, political instability, and economic insecurity, mothers contend with leaving children behind in pursuit of safety in another country. Mothers migrate expecting to settle in the United States (US) and send for their children in hopes of securing their collective safety over the long-term. Immigration policies, including those that separate families and limit avenues for asylum, influence and restrict decisions, actions, and opportunities available to migrating mothers. In the midst of complex structural and interpersonal dynamics, lived experiences of family and motherhood are altered and reshaped. This study offers a nuanced examination of transnational motherhood, grounded in the lived experiences of Central American women in the shadows of violence and migration to the US.

### 1.1. Transnational Motherhood

Transnational motherhood acknowledges the ways in which people are anchored to multiple spaces and processes of mothering span geopolitical boundaries. Mothering across borders involves ongoing emotional care of, provision of financial support to, and communication with children from afar [1]. The identities and roles of women evolve, span across, and transcend national borders, as do socio-emotional and economic dimensions of parenting. Hondagneu-Sotelo and Ávila’s seminal piece explored the meaning of motherhood in the context of immigration-related family separation [2]. Recent literature has resisted a binary conceptualization of transnational motherhood as either a sacrificial act or an abdication of responsibilities [3]. Contemporary scholarship suggests that mothering from afar is a prime example of how transnational identities function across social and physical spaces [4], expanding notions of gender and caretaking arrangements. Women who parent across borders face the basic question of “how to be socially and emotionally present while physically absent” [5] (p. 203). Rachel Hershberg uses the phrase “being present when forced to be absent” to describe how parents maintain and make meaning of transnational parenting [6]. Hershberg highlights consejos, the passing of conventional wisdom and advice from older to younger family members, as an example of adaptive communication between transnational parents and their children as a means of bridging distance and maintaining an emotional presence [6]. Horton describes transnational mothers feeling “physically absent but recognized and remembered in their home countries,” while also “physically present yet invisible and disavowed in the United States” [7] (p. 17).

Recent studies have investigated how family is reorganized through separation, fears of separation, and upon reunification [8,9]. Transnational motherhood research also explores the role of those who take on caretaking responsibilities when mothers migrate and cross-border communication about parenting decisions, discipline, and child-rearing expenses [3,5,10,11]. Parent–child relationships are not fixed; multiple variables, such as children’s age and developmental stage, communication, remittances, and the health and well-being of children, shape the experiences and outcomes of transnational motherhood [3,12]. Parents’ migration can positively affect their children’s economic well-being [11]. However, negative outcomes associated with migration-related separation include academic decline, ‘acting out’ behaviors, and poor self-esteem among children; and sadness, guilt, and loneliness among parents and caregivers [11,13,14,15]. Separation can also be associated with increased depression, anxiety, substance use, economic instability, and ambiguous loss for both children and parents [16]. Berger Cardoso and colleagues found that even when with their children, many parents feel a pervasive fear of separation and report that the risk of deportation, in addition to previous prolonged separations, alter family processes and family units [8].

### 1.2. Intersection of Migration and Violence

Women and girls in Central America are exposed to a constellation of violence, including from gendered impacts of criminal gang activity, to intimate partner violence (IPV), sexual violence, exploitation, and high rates of femicide [17,18]. Poverty, economic insecurity, underemployment, gender and age inequality, and patriarchal power structures intersect with risk of gender-based violence, placing women and girls at risk of experiencing abuse and violence and shrouding these experiences in silence with few outlets for support and protection [18]. Scholars describe migration as a strategy women employ to interrupt violence and ensure survival for themselves and their children [19,20,21,22]. Various forms of interpersonal, systemic, and structural violence intersect with the experience of migrating mothers. Non-partner sexual violence, IPV, and threats of violence, coercion, and abuse, for example, can make staying untenable [20]. Under these circumstances, women may migrate as an escape tactic or act of resistance [23,24]. The journey from Central America to the US, however, poses additional risks, during which women are exposed to verbal and physical abuse, sexual violence, exploitation, and other forms of violence [18,20,25]. Although studies describe the violence women face and their roles as mothers as embedded in the migration process from the beginning and playing critical roles in their decisions to migrate [20,21], motherhood is not widely addressed at the intersection of violence against women and migration.

### 1.3. The Current Analysis

The sensitive interplay between violence, migration, and motherhood represents an emerging and needed area of continued inquiry. This study draws from transnational feminisms, which recognize migration as a process involving complex, unsettled, and fluid senses of place, belonging, identity, and responsibility [10,26,27]. Feminist perspectives examine the many ways in which women assert power and resistance throughout migration, affirm women’s decisions, and address the multiple contexts, identities, and experiences shaping migration [23,28,29,30,31]. These perspectives amplify the experiences and effects of migration as extending beyond migrating individuals to children, families and extended networks, recognizing how economic responsibilities, parenting roles, and communication evolve across geographic boundaries [8]. Rooted in the critical transnational literature, this qualitative analysis examined processes of transnational motherhood at the intersection of migration and violence, grounded in the perspectives of Central American women over the course of their migrations to the US.

## 2. Materials and Methods

Themes of motherhood presented in this articled emerged inductively from a study conducted in 2014, which examined violence against women and processes of migration among Central American women in the US [20]. The study was approved by the University of Texas at Austin Institutional Review Board and adhered to ethical guidelines in researching violence against women and migration to ensure the self-determination, safety, and wellbeing of participants [32,33].

### 2.1. Recruitment

Based on a purposive sampling approach, researchers recruited women based on inclusion criteria: (1) 18 years of age and over; (2) migration to the US within the past 15 years; and (3) Central American country of origin: El Salvador, Guatemala, or Honduras. These criteria reflect the original study’s focus on contemporary patterns of migration from Central America to the US among adult women and women with children [20]. In recruiting potential participants, the first author drew on professional relationships with four organizations that provide shelter, social services, and/or legal services to immigrant communities in two cities in Texas. Staff presented the study to clients who matched the inclusion criteria and shared researchers’ contact information with women who expressed interest in participating.

### 2.2. Participants

Nineteen (*n* = 19) women participated in the study, eight of whom originated from Honduras, six from El Salvador, and five from Guatemala. The average age of participants was 35 years (range: 25 to 53 years). Their average length of time in the US was 5.4 years (range: 2 months to 15 years). All participants identified as mothers, and on average women had 3.26 children (range: 1 to 7). Seventeen participants experienced one or more forms of intimate partner violence, sexual violence, and/or human trafficking.

### 2.3. Data Collection

The lead researcher conducted in-person interviews, using an iterative approach to adapt questions and probes as data collection progressed. The semi-structured interview guide, which evolved over time, queried women’s motivations to migrate to the US and risks of violence prior to and during migration. Participants scheduled interviews at their preferred times and locations (e.g., the researcher’s office, participants’ homes, and service providers’ offices). In an effort to build rapport and develop trust, the lead researcher reminded potential participants that their legal immigration status was not a topic of inquiry and that researchers would not report immigration status to any governmental or other entity [32]. All participants gave verbal informed consent and received written information about the study in both Spanish and English. All participants received $25 cash at the beginning of the interview, along with clear reminders that interviews could be suspended or stopped at any time. At the conclusion of each interview, the researcher (a licensed social worker) shared resources related to any unmet social service needs shared during the interview. Interviews lasted between one to two hours. All interviews were conducted in Spanish and were digitally recorded and transcribed by a professional transcription service in Spanish. Data were stored in password-protected files accessible only to research staff.

### 2.4. Data Analysis

Informed by grounded theory analytical approaches [34], data analysis involved an ongoing examination of digital recordings, interview transcripts, field notes, and memos. The lead researcher first used line-by-line process coding to identify actions and processes across the first eight interview Spanish-language transcripts [34,35]. Next, she grouped these initial codes based on fit, frequency, or significance [34] and used these categories to code the full set of Spanish-language transcripts in NVivo. Participants’ reflections on motherhood vis à vis migration and violence emerged inductively from this analytical process. Iterative comparisons within and across participants resulted in the five processes described below. Researchers utilized several strategies to enhance rigor and trustworthiness: memo-writing, reflexive practice, maintaining an audit trail, peer debriefing, and thick description [36]. The primary researcher regularly debriefed and consulted with practitioners and researchers with relevant expertise.

## 3. Results

Motherhood was a salient and ever-present factor throughout women’s migrations from Central America to the US. The analysis identified five processes related to transnational motherhood at the intersection of migration and violence: risking everything, embodying separation, navigating reunification, mothering others, and experiencing motherhood due to sexual violence. All names are pseudonyms.

### 3.1. Risking Everything

Women’s efforts and risk-taking to protect and provide their children a life free from violence directly influenced their meaning making of motherhood. While some women migrated with one or more children, most women left their children in their home country when they migrated to the US. Women talked about fearing for their children’s safety back home and feeling guilt for having left them. María stated her hopes for her children: “tienen derecho a vivir un futuro sin violencia, sin problemas, sin amenazas de muerte.” (*“They have a right to live a future without violence, without problems, without death threats.”*) Isabel also described her efforts to break the cycle of violence she had experienced, “no quería que mi historia se repitiera en mis hijos y por eso yo he sido bien luchadora.” (*“I didn’t my kids to repeat my history and that’s why I’ve been a fighter.”*) Sierra was in an untenable position as a mother trying secure safety for herself and her two young children in the context of regional gang violence in Honduras. She had only enough money to pay for herself and one child to travel to the US. On the one hand, she was concerned about the gangs that controlled her community targeting her son; on the other hand, she had to consider the likelihood that her daughter would be at risk of sexual assault and exploitation if she remained in Honduras. Sierra weighed the risks and ultimately decided to bring her daughter to protect her from sexual violence, leaving her son in Honduras. María first migrated from Guatemala to the US without her children. She then returned to Guatemala, hoping to remain there with her family. However, her life was again threatened and she decided to make the journey again, this time with her children, which she and her mother both perceived as a decision to risk everything. With great emotion María shared,

Agarré una mochila y metí la ropa que pude de mis niños, y otra bolsa la llené con comida. Y [mi mamá me dijo], No. Estás loca, vas a ir a matar a tus hijos, eres una asesina. Le dije no, no lo voy a hacer, pues si así pasa prefiero que muramos todos. (llorando) Le dije, no me importa, lo único que me importa es tratar de hacer algo por ellos, que no un día me voy a arrepentir y dije lo pude hacer y no lo hice. Y quería escapar, de ese infierno.


*I grabbed a backpack and filled it as much as I could with my children’s clothes and another bag I filled with food. [My mother told me], “No, you’re crazy, you’re going to get your children killed. You’re a murderer.” I said, “No, I’m not going to do that, and if that does happen I would rather we all die.” (crying) I said, “I don’t care, the only thing that matters is that I try to do something for them, I don’t want to one day regret being able to do something and not doing anything. I want to escape from that hell.*


For María, taking leave from her mother in Guatemala also included a desire for her mother’s permission and blessing, and she told her mother, “yo solo quiero su bendición es todo lo que le pido.” (*“I just want your blessing, that’s all I’m asking of you.”*) Part of “risking everything” included the possibility that the decision to migrate could damage relationships with family at a time when they needed their loved one’s blessings and emotional support to undertake a dangerous trip. The decision to migrate put the support and trust from their families and their most important relationships at risk.

### 3.2. Embodying Separation

When separated, mothers constantly thought about their children and worried about their safety, well-being, and schooling. Mothers spoke at length about the stress and sense of guilt related to arranging care across borders. They expressed concerns about how to send money, and whether it would arrive and adequately cover their children’s needs. Despite feeling some satisfaction with being able to provide for their children’s education or health needs, women felt distress at not being physically present to keep their kids safe from gang violence. Women experienced the separation from their children in terms of acute suffering and fear. Sierra recalled leaving her son in Honduras, “mi corazón se partió en dos, dejar la mitad allá y traerme mi otra mitad para aquí.” (“*My heart broke in half, leaving one piece over there and bringing the other with me here.”*) After talking with her children during her travels through Mexico, Clara said, “ponía a llorar después de oír a mi hijo. Y decía, ojalá que esto valga la pena.” (“I would start crying after hearing my son. I would say, ‘I hope this is worth it.’”) Beatriz referred to these worries as, “las madres rompen la cabeza pensando.” (“*mothers worry themselves sick.”)* In addition to fear and concern, women felt a tremendous sense of guilt in having migrated and the resulting separation from their children. Beatriz, for example, returned to Guatemala after four years in the US, fulfilling a promise she had made to her children. When she arrived back in Guatemala, they passed by her in the airport without recognizing her. She said,

Me sentía tan culpable haberlos abandonados cuatro años. El pequeño ya no me reconocía. Tuve que pegarme mucho a él para que me volviera a reconocer. Tuve que ganarme el cariño de los tres nuevamente, y pedirles perdón.


*I felt so guilty after leaving them abandoned for four years. The smallest didn’t recognize me. I had to stay close to him a lot so that he would remember me again. I had to win the affection of the three all over again and ask them for forgiveness.*


Women shared painful examples of how motherhood plays out in separation from home and children throughout migratory processes. While Celia migrated to the US with one child at great emotional and physical cost, she continued to grieve the death of her son, who had died before she left Honduras. Her physical distance from Honduras played a significant role in her grief. She was plagued with not being in Honduras to provide him a proper burial. She was no longer in contact with anyone who had known him and felt that she, alone, carried his memory. Furthermore, she carried the burden of her abusive husband blaming her for the boy’s death. Hortensia shared another painful example of embodying separation during migration. While being held against her will in a drop house and having been sexually assaulted by smugglers, Hortensia learned that gang members had murdered her oldest daughter on the street in Guatemala. Hortensia was later able to bring two of her four surviving children to the US, though she remained physically separated from her two children in Guatemala and permanently separated from her deceased daughter. In spite of the separation, however, women described their children and motherhood as a source of strength. María, for example, shared “por mis hijos, no me da miedo nada.” (“*because of my children, I’m not scared of anything.*”) Women drew strength from having migrated for their children and felt supported by their children’s love and encouragement from afar. Facing inconsistency of support systems over time and space, women described ongoing processes of losing support, building support, and re-building supportive relationships with their children and others.

### 3.3. Braving Reunification

All women in the study eventually brought some or all of their children to the US or were in the process of navigating reunifications with their children. Despite having survived harrowing journeys themselves, several women relied on unofficial means of bringing their children to live with them in the US. This involved paying a smuggler to bring them through Mexico and across the border, and exposing children to similar risks they endured while migrating, including exposure to the elements, lack of food and shelter, and vulnerability to robbery, violence, and exploitation. Isabel, for example, described sending for her children, losing track of them for an anxiety-ridden week, and being extorted by the criminal gang network that controlled border crossings. She shared,

Yo puse en riesgo la vida de mis hijos, me los tuvieron ahí una semana exacta, ocho días sin saber nada, yo me estaba muriendo porque sentía que ¿dónde los iba a hallar? México también es grande, ¿dónde los iba a encontrar? Yo decía que si yo no sabía de ellos, yo me iba a ir a buscarlos de lugar en lugar, que yo los tenía que encontrar.


*I put my children’s lives in danger. [The smugglers] had them there for exactly one week, eight days without knowing anything. I was dying because I felt like Mexico is so big. Where was I going to find them? I said that if I didn’t hear anything about them I would go place by place looking for them, I had to find them.*


The smugglers demanded Isabel send more money, ultimately totaling almost $20,000, which she did in desperation to secure the safety of her children, pleading to see them again. The smuggler said to her, “sí te los voy a entregar, cierto que eres una buena persona, una buena mamá, porque a pesar de que no sabes nada de ellos, sigues insistiendo” (“*Yes, I will give them to you. You are a good mother because despite not knowing anything about them, you kept insisting.”*) The US border patrol finally located them and sent them to the hielera [“*freezer*” or “*ice box*” in reference to the temporary detention facility] where they spent three days. From there, US immigration officials sent them to a center for minors where Isabel went to get them. Karla had two children in Honduras and gave birth to a third child in the US. She and her husband worked hard to save money to bring the older children to the US. They twice collected the required money, only to have it stolen. The third time they paid $20,000, with the intention of bringing both boys safely, avoiding the danger and suffering they had experienced. Yet, despite promises of better conditions, the boys ultimately travelled for one month by train, slept outside, and went without food. After six years of separation, Karla described their reunification: “Estaban enojados con nosotros, porque dijeron cómo era posible que nosotros habíamos mandado traerles así. (“*They were angry. They said how was it possible that we would send for them to be brought like that?*”) Some women reported that they would never bring their children to the US unless they could do it by legal means. Natalia, for example, was determined not to subject her other children to the abuse she and her daughter had endured during the journey. She reflected,

Es algo que a mi me llena de tristeza, y por eso es que yo ya no me animé a traer a mis hijos así, a mis otros dos hijos. Yo dije no, ya no más. Porque si mi niño no camina, o si a mi niña le pasa algo, y lo van a dejar botado, yo digo que no ya no.


*It’s something that fills me with sadness, and that’s why I couldn’t convince myself to bring my other children. I said no, because if my boy couldn’t keep up with the walking, they would abandon him, or if something happened to my girl. I say no, no more.*


In two cases, women described how IPV complicated the process of reuniting with children. Once in the US, Karla’s husband, became jealous, controlling, and abusive, which ultimately affected her ability to work and hindered their ability to save for their sons’ travel to the US. Alma shared similar difficulties even though with a T Visa she had a legal avenue through which to bring her children. Her abusive ex-partner managed to thwart Alma’s efforts to get the children’s documentation and at the time of the interview, she had not yet been able to bring any of her four children to the US.

### 3.4. Mothering Others

Women’s experiences with transnational motherhood also involved taking care of unaccompanied children they encountered along their journeys, particularly through Mexico. In the midst of missing their own children and maternal roles, women sometimes found themselves mothering others in transit. María, for example, helped care for eight-year-old twins during the entire journey through Mexico. Lorena spent several days taking care of a ten-year-old girl who had traveled alone from Guatemala, whom she encountered at a hotel. She shared,

Nunca le pregunté cómo se llamaba, porque la niña estaba callada y no hablaba. La encontramos ahí en el hotel y yo la agarré como que era mi hija, como que traía acá. La bañé y la peiné. La regañe porque no se quería poner sweater. Lavé sus calzones, ‘vete a cepillar los dientes’ y así pasamos la noche. Nos subimos a otro bus y fuimos a otro pueblo. Y como la niña se pegó un poquito conmigo, entonces se quedó conmigo.


*I never asked what her name was because the girl was quiet and didn’t talk. We found her there in the hotel and I grabbed her like she was my daughter, as if I had brought her there. I bathed her and combed her hair. I scolded her because she didn’t want to put on a sweater. I washed her underwear, [told her to] brush her teeth, and that’s how we spent the night. We got another bus and went to another town. The girl got a little attached to me, so she stayed with me.*


Sierra travelled with her own child and took care of two other children who were travelling alone and hoping to reunite with their respective mothers in the US. To protect them from harm, she told people that they were her own children and her travel group did not realize they were not her children until after they had crossed the Rio Grande. While women cared for unaccompanied children as if they were their own, these relationships were temporary. During or shortly after crossing into the US, they separated. Lorena continued to worry about what happened to the child she took care of in Mexico, who was passed to other smugglers. She questioned, “¿Y porque va a estar mejor que contigo si a ti no te conoce? A mi tampoco pero ya tiene dos o tres días de estar conmigo, pero a ti ni una vez te ha visto.” (“*Why is she going to be better off with you if she doesn’t know you? She doesn’t know me either but she’s spent two or three days with me and she’s never even seen you.*”) Women took advantage of opportunities to care for other women’s children, putting their strengths and skills to use while separated from their own children and in the absence of unaccompanied children’s mothers.

### 3.5. Experiencing Motherhood due to Sexual Violence

Complicating the meanings women made of motherhood were experiences of sexual violence and resulting pregnancies. Three women in the sample disclosed (unprompted) that they had experienced or witnessed pregnancies following rape prior to, during, or post migration. Beatriz, for example, became pregnant resulting from sexual abuse by the man (“coyote” or smuggler) helping her cross through Mexico and into the United States. Lorena described immigration officials making women take a pregnancy test after they were apprehended while crossing the border into the US. She went on to say,

Casi la mayoría de muchachas que vienen llegando y tienen sus hijos aquí fueron violadas en el camino y el bebé es consecuencia del camino. Es traumante. Porque aparte de que pasa esas penas en el camino todavía te queda un recuerdo para toda tu vida. (llorando)


*Almost all the women who arrive and have their kids here were raped on the way, and the baby is a consequence of the trip. It is traumatic, because in addition to experiencing these challenges on the journey, you are left with a memory of it for your entire life. (crying)*


Gloria, a mother of seven children, described what became a lengthy period of commercial sexual exploitation upon her arrival in the US after fleeing IPV in Honduras,

Cuando llegamos aquí nos amarraban de las manos y de los pies y un pañuelo en la boca y nos tiraban a una camioneta como una pelota. Cuando llegamos a una casa, nos compraban una ropa y nos obligaban a que nos vistieramos y traían una señora que los peinara y los pintara. Y nos obligaban a cosas que no queríamos. Y eso fue duro para mi. Pues mis hijos son de todo eso. Los tres hijos de aquí fueron del abuso.


*When we got here they tied our hands and feet up, a handkerchief in our mouths and they threw us in a truck like a ball. When we got to a house, they bought us clothes and they made us put it on and there was a woman who brushed our hair and did our make-up. And they made us do things we didn’t want to do. That was difficult for me. My children are from all of that, the three children here are from the abuse.*


The spontaneous disclosures of these three women about the ways that violence forever marked motherhood in complex ways provide an additional and significant facet of transnational motherhood, deserving of further attention.

## 4. Discussion

The findings revealed the ways in which women who had migrated from Central America to the U.S. made meaning of their identities and roles as mothers, highlighting the centrality, power, and fluidity of motherhood at the intersection of migration and violence. Their roles as mothers and the imperative to risk everything to secure safety for themselves and their families served as constants in shaping their internal motivations and decisions to act. Women endured multiple separations from their children to secure safety, and braved the implications of reunifying with children who did not accompany them in their original journeys. In transit, they stepped in to mother children who were making the journey without a parent, fulfilling their roles as mothers even if only in passing. Some women would not make it to the U.S. without being sexually assaulted and experiencing motherhood as a direct result of violence. Together, the themes provide insights into complex processes shaping notions of motherhood and inform next steps for research, policy, and community practice.

Motherhood intricately intertwined with the violence-migration nexus, as women sought to protect their children from harm and create a violence-free future for their children and themselves. Experiences and possibilities of violence and migration rendered transnational mothering fraught and heavy with impossible decisions. In fleeing violence both at home from an intimate partner and in the public sphere (e.g., gangs), they confronted additional threats and assaults, as well as continued structural violence. A dominant factor in a mother’s migration is often to ensure the economic survival of the family, which is largely interconnected with gendered poverty and structural oppression as well as the interpersonal economic coercion and control common in IPV [20]. Thus, any positive migration-related economic opportunities were still constrained by financial obligations back home and by IPV-related economic coercion persisting across borders and experienced anew in the US.

The findings from the current study add to the growing migration-related literature by highlighting the effects of family separation on mothers. The effects of temporary and enduring separations have profound and varied effects on parents, children, and caregivers [8,13,14,15,37]. What may be considered a temporary separation from children may become lengthy or even permanent, and women may be unprepared for the length of separation and the toll it takes on parent–child relationships [10]. Mothers interviewed for this study spoke at length about the stress and sense of guilt related to arranging care across borders and the social and emotional isolation and loss of support and trust from family members. Communication between mothers and children, as well as communication between mothers and their children’s caregivers, is crucial to exchanging information, making parenting decisions, and maintaining emotional connections [5]. Women recognize the role of caretakers (primarily their children’s maternal grandmothers) as critical in supporting their migration to the U.S. [11]. While some argue that leaving children behind is traumatic to children and contributes to family disintegration, others contend that parent substitutes are fully capable of taking on caretaking responsibilities [3]. Perhaps most impactful were the stress and trauma associated with the knowledge that their children continued to face danger back home, and the enduring and ambiguous losses associated with the death of a child or children. At the same time, women shared the positive impact of transnational mothering in the sense of agency and action incurred by doing what they had to do in the face of untenable circumstances and by taking advantage of opportunities to care for unaccompanied children in solidarity with other mothers facing similar circumstances.

The findings must be considered in the context of US immigration, detention, and deportation policies and practices, which directly influence and often manufacture family separation and exacerbate experiences of IPV. Contemporary policies, for example, have extended the duration of and uncertainty around family separation, restricted cross-border travel for transnational caretaking, and generally have made family reunification more difficult [5,16,20,38]. Menjívar argues that US immigration law exacerbates long and indefinite separations [39]. Family separation came to the foreground with the zero tolerance policy announced in May 2018, during which US officials tragically and haphazardly separated thousands of children from their parents [40]. Other recent policies, such as the Remain in Mexico Program (officially titled Migrant Protection Protocols), further restrict and hinder migrating parents’ parenting decisions and actions. Under this practice, more than tens of thousands of asylum-seekers were detained along the US-Mexico border and forced back into Mexico to apply for asylum [41]. Waiting in makeshift camps, exposed to the elements and a host of other gendered dangers, many waited for months for their claims to be processed. Those travelling with children parented under these conditions and amidst great uncertainty, often unable to move neither north or south, and with few opportunities to create stability or safety for their children. While media coverage and public outrage over the zero tolerance policy and Remain in Mexico were merited, they overshadowed manifestations of family separation that occurred and continue to occur both as a common component of forced migration and as long-standing US government practice. The US system of immigration detention further strains mothering and mother-child relationships, particularly among those who have experienced IPV [42].

It is important that conceptualizations of transnational mothering acknowledge the power women exert as mothers. As this study illuminates, migrating with children and migration-related separation from children involves fluid and interconnected forms of coercion and/or force but also encompasses choice, autonomy, and self-determination. Even in the midst of the aforementioned policies and practices that restrict and criminalize the human experience of parenting, this study highlights the multiple ways in which women resist and challenge conditions and strategies that threaten to limit and constrict their roles as mothers. In warning of the dangers of maternal narratives that so easily blame mothers, rob women of agency, and promote one-dimensional and damaging narratives, Geraldine Pratt suggests that scholars instead aim to present “responses that affirm the testifier’s capacity to respond and hence their agency and subjectivity” [43] (p. 10). These findings support Pratt’s cautionary reminder about the responsibilities for scholars and researchers and extend to the ethical obligations of social workers, community organizers, and mental health practitioners. Efforts to respond in practice must identify, support, and expand women’s agency and power as transnational mothers. Social workers, community organizers, and mental health providers working with immigrant communities must also work to address internalized guilt and shame and the larger social mechanisms that promote harmful narratives.

It is important to note certain limitations associated with this study, as it explored the process of migration in the context of violence for a narrow snapshot of migrating women and cannot represent all Central American women. Any research that focuses on migrating women runs a risk of essentializing migrant women and mothers, in particular, and it remains crucial to recognize the heterogeneity of migrating women and of migrants in general. In addition, researchers faced common impediments with recruiting, which were likely due to the social isolation and climate of fear experienced by undocumented and precariously documented immigrants, in addition to participant and resource constraints that did not allow for multiple interviews with each participant nor member-checking. There also exists potential bias related to participants’ prior engagement with legal immigration systems and having already told, constructed, or re-constructed their stories for a different audience [44,45,46]. Sampling and recruitment procedures may have also resulted in bias towards those who had sought services, which could have included those who had faced more intense experiences of violence or hardship. Furthermore, while this study explored motherhood at the intersection of migration and violence, other elements of migrating parents’ identities and experiences are missing. Not all migrant women share the same identities in terms of gender and sexual orientation, roles as mothers, or socioeconomic status, and this study only interviewed those identifying as women, excluding girls, men, and the transgender community. Further analysis and research should explore the ways in which racial, ethnic and indigenous social positions shape motherhood in the context of violence and migration, and investigate the parenting experiences of fathers and gender non-binary parents as well. Finally, this study did not directly or systematically investigate the critical issue of violence-related pregnancies, an aspect of transnational motherhood meriting further attention.

## 5. Conclusions

The findings from this analysis highlighted facets of transnational motherhood at the intersection of violence and migration, amplifying the perspectives and experiences of mothers as they “risked everything” to secure a safe future for themselves and their children. Transnational motherhood encompasses a multitude of parenting configurations, as mothers continue to parent from afar and to reconfigure their caretaking pre-migration, during migration, and after reunification. Centering the voices, perspectives, and experiences of women who embark on the migration journey to fight for themselves and their children is imperative to research, practice, and advocacy to address structural violence spurring migrations and to change oppressive immigration policies. The study reiterates the importance of understanding how violence shapes and informs women’s migrations and decision-making, and the consequences women endure in taking action to change the inevitability of violence in their own and their children’s lives.

## Data Availability

Not applicable.

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
