# Peer review of "“Mi Corazón se Partió en Dos”: Transnational Motherhood at the Intersection of Migration and Violence"

_ijerph, 2022, doi:10.3390/ijerph192013404_

Round 1

Reviewer 1 Report

The article is an excellent example of insightful and extraordinarily well written research utilizing grounded theory. It is clearly structured, very well referenced, and remarkably concise, while providing very illustrative examples making a strong case for their general argument. The grounded theory approach fits perfectly to your research interest, the research question and the data you analyse. Congratulation for this extraordinary article.

There are only very few issues, which I want to bring to your attention and which need revision:

Section 2.1 Recruitment

You indicate that you interviewed women, 18 years of age and older, who migrated within the past 15 years from Guatemala, El Salvador or Honduras. However, you do not provide an explanation for those decisions. What is the rationale behind interviewing women from those three countries and not from others? Was it coincidence that women were from those three countries, or is there a particular reason for focusing on women from those three countries in particular? The same goes for age and migration history. I don't think that this has to be adressed very detailed, but some explanation for those decisions would be good.

Section 3.4 Mothering Others

While I follow you here for the most part, I am not yet convinced from what you write in line 318. You state that "in missing their own children and maternal roles, [the women] turned to mothering others in transit." What reasons do you have to interpret the women's action as being a result of them missing their children? In your discussion (line 422-426) you emphasise that there might be a sense of solidarity involved behind the "mothering of others". To me, this interpretation appears more in line with the examples you presentyour paper. If you really think that it is also a kind of action resulting from them missing their own children, you should provide an example from your data, which illustrates and backs up this interpretation.

section 4 Discussion

Overall the discussion is very well written (particulary the fifth paragraph makes a very strong and lucid argument). However, I would appreciate one or two sources regarding the potential biases, especially the second point you mention (line 471-473, "prior engagement with legal immigration systems and having already told, constructed, or re-constructed their stories for a different audience"). Here works on narrative research and the effect of audiences on story telling (e.g. Catherine Kohler Riessman, Narrativ Methods for the Human Sciences, SAGE, 2008), or empirical research on migrant and refugee's story telling might be helpful.

Line 210-212 This sentence is slightly unclear - what do you mean with the inserted "and part of 'risking everything' "? Is "the decision to migrate" in itself part of "risiking everything"? Or are the "fears of damaging relationships..." part of 'risking everything'? I think you should clarify this sentence.

Line 250-253 This sentence needs to be revised. I think there's a verb missing in the second part; and it appears to me, that the "both" should not be there.

line 353 It appears to me, that the "coyote" in brackets serves to show how Beatriz called the men, hence I would put it under quotation marks to clarify this.

line 486 I would put quotation marks around "risked everything", as I would not subscribe to the interpretation that the women risked - literally - "everything".

In sum, I want to congratulate you for this extraordinary article!

Reviewer 2 Report

This is an extremely interesting work that presents the results of interviews with a highly vulnerable and hard-to-reach target group. However, the scientific presentation and synthesis of the material, theory and further conclusions is immature. In view of this, I recommend the authors to use the possibilities of the journal and add a comprehensive description of the results, as well as still invest time in the explanations and merging of results and theory, so that the conclusions can be implemented in a comprehensible and theory-based way using the material. Concerning the method, the limitations include essential aspects, but it needs further elaboration due to the high vulnerability of the target group. Among other things, it is very disadvantageous that interviews were conducted once, whereas it is recommended to meet interviewees several times and to build up a real relationship of trust. Furthermore, the possibility to share articles with the interviewees before publication can be used, e.g. to be able to set additional steps of anonymization if necessary.
The applied method of grounded theory requires a strong theoretical reference, which is not nearly sufficiently given or has not been presented. Here, too, there is a strong need for revision and a more comprehensive elaboration of applied theories and their merging with the results.

Round 2

Reviewer 2 Report

Thank you for the revised version of your manuscript and replies to comments. As already mentioned in the first comments, I think the study is very valuable in the context of the target group and the extremely relevant research questions. Therefore, after reviewing the revised version and responses to comments, I recommend publication. I hope that more studies will emerge on this complex and highly sensitive topic.
